# Concept-Level Explainability for Auditing & Steering LLM Responses

<p align="center">⚠ THIS PAPER CONTAINS MODEL-GENERATED CONTENT THAT MIGHT BE OFFENSIVE. ⚠</p>

## ABSTRACT

As large language models (LLMs) become widely deployed, concerns about their safety and alignment grow. An approach to steer LLM behavior, such as mitigating biases or defending against jailbreaks, is to identify which parts of a prompt influence specific aspects of the model's output. Token-level attribution methods offer a promising solution, but still struggle in text generation, explaining the presence of each token in the output separately, rather than the underlying semantics of the entire LLM response. We introduce *ConceptX*, a model-agnostic, concept-level explainability method that identifies the *concepts*, i.e., semantically rich tokens in the prompt, and assigns them importance based on outputs' semantic similarity. Unlike current token-level methods, ConceptX also offers to preserve context integrity through in-place token replacements and supports flexible explanation goals, e.g., gender bias. ConceptX enables both *auditing*, by uncovering sources of bias, and *steering*, by modifying prompts to shift the sentiment or reduce the harmfulness of LLM responses, without requiring retraining. Across three LLMs, ConceptX outperforms token-level methods like TokenSHAP in both faithfulness and human alignment. Steering tasks boost sentiment shift by $0.252$ versus $0.131$ for random edits and lower attack success rates from $0.463$ to $0.242$, outperforming attribution and paraphrasing baselines. While prompt engineering and self-explaining methods sometimes yield safer responses, ConceptX offers a transparent and faithful alternative for improving LLM safety and alignment. Beyond demonstrating the practical benefits of attribution-based explainability in guiding LLM behavior, this work introduces *steering effectiveness* as a novel measure of XAI quality. [1].

## 1 INTRODUCTION

Large language models (LLMs) are widely used in real-world applications, such as conversational agents (OpenAI, 2024a), but concerns remain about their safety and alignment with human values (Wach et al., 2023; Ji et al., 2023; Wei et al., 2023; Hazell, 2023). Despite efforts to align models (Ouyang et al., 2022; Bai et al., 2022; Korbak et al., 2023), LLMs still generate harmful or misleading content due to flawed training or adversarial attacks (Ma et al., 2025; Chen & Shu, 2023; Spitale et al., 2023; Mouton et al., 2024; Wan et al., 2024; Fang et al., 2024). Such misalignment can emerge from malicious fine-tuning (Betley et al., 2025) or adversarial prompts that bypass safety defenses (Zou et al., 2023; Meinke et al., 2024).

Attribution-based explainability methods offer a promising approach to identifying input elements that lead to harmful or biased outputs from LLMs (Wu et al., 2024). While effective in classification settings, these methods face challenges in text generation due to the open-ended nature and semantic variability of responses. Existing approaches typically operate at the token level, measuring importance based on the likelihood of reproducing specific output tokens (Goldshmidt & Horovicz, 2024; Amara et al., 2024). This leads to three major limitations: (i) their objective is on literal token overlap rather than semantic meaning, failing to capture paraphrased and semantically equivalent responses (Wu et al., 2024); (ii) they overlook concept sensitivity, often focusing on uninformative function words (e.g., "the", "is"), whilst effective XAI requires both token- and concept-level perspectives; and (iii) they treat tokens as independent features, which breaks the contextual coherence necessary for meaningful text, resulting in misleading attributions when tokens are isolated (Vadlapati, 2023; Chen et al., 2020b).

To overcome these challenges, we propose **ConceptX**, a family of concept-level, attribution-based explainability methods. Built upon a coalition-based Shapley framework, ConceptX addresses the three current limitations. First,

---

[1]The code is anonymously available at `https://anonymous.4open.science/r/ConceptX`

<p align="center">1</p>

instead of optimizing for token-level reproduction, it uses a semantic similarity objective, ensuring that concept attributions reflect changes in meaning rather than sticking to the form of the output. Second, it focuses on input *concepts*, i.e., semantically rich content words from ConceptNet (Speer et al., 2017), better suited for concept-aligned LLMs and yielding more interpretable, actionable explanations. Third, ConceptX evaluates input concepts in context while preserving the sentence's grammatical and semantic structure during attribution. It does so by introducing two alternative concept replacement strategies alongside traditional removal. Thanks to its similarity-based optimization, ConceptX can generate aspect-specific explanations by identifying what input concepts drive a particular semantic dimension of the output, beyond simply reproducing the original response. This allows users to audit and address the causes of undesired model behaviors. With this capability, ConceptX becomes a powerful tool for targeted prompt-level interventions: by detecting influential input concepts, users can steer LLM outputs without requiring retraining or fine-tuning. This makes ConceptX a lightweight yet effective approach for advancing both explainability and alignment in LLMs.

Our model-based evaluation on the Alpaca dataset (Taori et al., 2023) shows that ConceptX provides more faithful explanations than prior attribution methods like TokenSHAP (Goldshmidt & Horovicz, 2024). In addition, we show that ConceptX can be used for both **auditing** and **steering** the text generation process. In particular, the human-based evaluation of our designed GenderBias dataset shows ConceptX's effectiveness in identifying semantically meaningful drivers of biased outputs. Results are consistent across three LLMs and suggest that ConceptX can be used for **auditing** LLMs by generating concept-level attributions and optimizing them for similarity to target aspects (e.g., bias or harm). Beyond explanation, ConceptX attributions can also guide prompt-level interventions by identifying which input concepts to modify for **steering** LLM outputs. We demonstrate this in two use cases: sentiment polarization, where ConceptX more effectively shifts sentiment than TokenSHAP, and jailbreak defense, where it reduces attack success and response harmfulness better than attribution and paraphrasing baselines (Goldshmidt & Horovicz, 2024; Cao et al., 2023). While generative and prompt-based methods remain stronger in harm mitigation, they also come with the computational and annotation overhead of fine-tuning and prompt engineering. In contrast, ConceptX offers a lightweight, interpretable, and actionable alternative for guiding LLM behavior. Our contributions can be summarized as follows.

- We introduce ConceptX, a family of concept-level attribution methods that addresses key challenges in text generation explainable AI (XAI) by focusing on semantics and enabling aspect-targeted explanations.
- We demonstrate that ConceptX generates more faithful and human-aligned explanations when auditing LLM outputs compared to current model-agnostic attribution methods.
- We propose a prompt-level steering method using ConceptX to edit aspect-relevant concepts, showing superior performance in mitigating sentiment and harmfulness.

Through applications in bias, sentiment, and harmful content, ConceptX demonstrates how explainability can directly support alignment, with steering effectiveness serving as a practical metric for explanation quality.

## 2 RELATED WORK

**Attribution Explainability Methods in NLP.** LLM explainability seeks to identify the underlying reasons behind a model's outputs, such as harmful content or specific target aspects, providing a foundation for more effective intervention. Common attribution methods developed for traditional deep models include gradient-based methods, perturbation-based methods, surrogate methods, and decomposition methods (Murdoch et al., 2019; Du et al., 2019). In NLP, the most prominent XAI techniques include feature importance and surrogate models (Danilevsky et al., 2020). These methods may focus on different explanation targets, such as word embeddings, internal operations, or final outputs, leading to a division between model-specific and model-agnostic approaches (Zini & Awad, 2022). Mechanistic interpretability focuses on internal model mechanisms, examining activation patterns and neuron roles (Vijayakumar, 2022; Sajjad et al., 2022), whereas model-agnostic attribution methods assign importance scores to input features (typically tokens) based on their influence on the model's prediction. Built on general-purpose techniques like SHAP (Shapley et al., 1953) and LIME (Ribeiro et al., 2016), those attribution methods have been adapted for text data to account for syntactic constraints and word dependencies (Amara et al., 2024). Although traditionally applied to classification tasks (Kokalj et al., 2021; Chen et al., 2020a), recent work has extended these methods to autoregressive models, aiming to shed light on the generative processes of language models (Amara et al., 2024; Goldshmidt & Horovicz, 2024). In this paper, we introduce a model-agnostic, concept-level explainability method that identifies semantically rich tokens in the prompt and assigns them importance based on the outputs' semantic similarity.

**Leveraging Explainability for LLM Alignment.** As LLMs grow more powerful, their lack of explainability poses serious ethical risks, undermining efforts to detect or mitigate harms like bias, misinformation, and manipulation. XAI techniques are thus crucial for auditing and aligning these models with human values (Hubinger et al., 2024;

Zhao et al., 2024; Martin, 2023). For example, data attribution tools and attention visualizations can expose biases such as gender stereotypes (Li et al., 2023c), while probing classifiers help identify harmful associations embedded in model representations (Waldis et al., 2025). Attribution-based explanations can serve as indicators to detect LLM hallucinations (Wu et al., 2024). However, integrating explainability to AI alignment also comes with challenges: neural networks remain difficult to fully understand (Elhage et al., 2022), and unaligned AIs may even develop incentives to evade interpretability tools (Benson-Tilsen & Soares, 2016; Sharkey, 2022). Coalition-based methods like ConceptX offer model-agnostic explanations of how input semantics shape outputs, circumventing LLM evasion strategies, in order to discover possible reasons for harmful or biased responses.

**LLM Steering and Defense Methods.** To defend against malicious use and align LLMs with human values, researchers have developed a range of steering and defense methods that intervene at different levels: input, prompt, or internal model representations (Ma et al., 2025). Input-level approaches include perturbation and paraphrasing techniques (Cao et al., 2023; Robey et al., 2023; Yan et al., 2023), token filtering (Wang et al., 2024a; Liu et al., 2024), translation-based back-translations (Wang et al., 2024b), and attribution or detection strategies using gradients, attention scores, or perplexity (He et al., 2023; Li et al., 2023a), LLM self-defense (Phute et al., 2023). Prompt engineering methods such as SafePrompt (Deng et al., 2023) and Self-Reminder (Xie et al., 2023) shape outputs by embedding behavioral constraints or reformulating queries. Internal steering techniques include activation steering, which manipulates intermediate representations to shift model behavior (Gao et al., 2024; Li et al., 2023b), and sparse autoencoder (SAE)-based approaches that identify and control interpretable features in activation space (Bricken et al., 2023; Cunningham et al., 2023). Although not yet widely applied to LLM alignment, attribution-based explainability methods could enhance input-level steering by directing perturbations toward the most influential input features.

## 3 METHOD

### 3.1 OVERVIEW

ConceptX introduces a concept-level coalition-based attribution approach. The objective is to discover the *semantic contribution* of input concepts to a target text. In contrast to prior Shapley-based methods for textual data, such as TokenSHAP (Goldshmidt & Horovicz, 2024) and SyntaxSHAP (Amara et al., 2024), which operate at the token level, ConceptX targets only semantically rich units by excluding function words and low-information tokens. Those units referred to as **concepts** correspond to content words with high semantic value, quantified using their node degree in the ConceptNet knowledge graph (Speer et al., 2017). ConceptX's methodology consists of two main stages: *concept extraction* and *concept importance estimation*. During the concept extraction, key input concepts are identified using a content word extraction and the knowledge graph ConceptNet (Speer et al., 2017)'s connectivity. Then, ConceptX uses a Shapley-inspired Monte Carlo strategy (Goldshmidt & Horovicz, 2024) to estimate the influence of each concept on a specific explanation target. When estimating concept coalitions, ConceptX replaces unselected concepts following three strategies: *r*emoving the concept (*r*), replacing it with contextually *n*eutral alternatives (*n*), or an *a*ntonym (*a*). Replacing instead of omitting (Goldshmidt & Horovicz, 2024) preserves grammatical correctness. Neutral or antonym replacements maintain linguistic coherence while altering the semantic content, allowing us to isolate the semantic influence of concepts. Cosine similarity between the explanation target – initial LLM **B**ase output (**B**), **R**eference text (**R**), or **A**spect (**A**) – and the modified outputs serves as a value function to estimate concept importance. An aspect refers to a specific semantic property or quality expressed in a sentence, such as sentiment (e.g., positive or negative), bias, toxicity, or safety. Figure 1 illustrates the different steps in the case of neutral replacement.

**Notations.** Throughout the rest of the paper, we use the notation ConceptX$_{\text{TARGET}}$-*repl.strat.*, where the subscript denotes the explanation target (B, R, or A) and the final italic letter specifies the concept replacement strategy used to evaluate coalitions (*r*, *n*, or *a*). This convention allows us to isolate the impact of each methodological variation. For example, ConceptX$_A$-*n* refers to the variant using neutral concept replacement and an aspect-based value function. Refer to subsection B.1 for a list of all method combinations. Unless stated otherwise, *ConceptX* refers to the full set of such method combinations.

**Replacement Strategy**

*r* = *r*emove
*n* = *n*eutral replace
*a* = *a*ntonym replace

**ConceptX$_B$-*n***

**Explanation Target**

B = LLM **B**ase response
R = **R**eference text
A = **A**spect

### 3.2 CONCEPTS AS INPUT FEATURES

The first step in ConceptX is to extract the concepts that will serve as input features and receive importance scores. Unlike Shapley-based text methods, ConceptX ignores function tokens (e.g., prepositions, articles, conjunctions), focusing instead on content words (nouns, verbs, adjectives, adverbs) to provide faithful and human-interpretable explanations. Concepts are matched to entries in the ConceptNet (Speer et al., 2017), a knowledge graph with over 8 million nodes and 21 million edges, where semantic richness is measured by node degree. Extraction

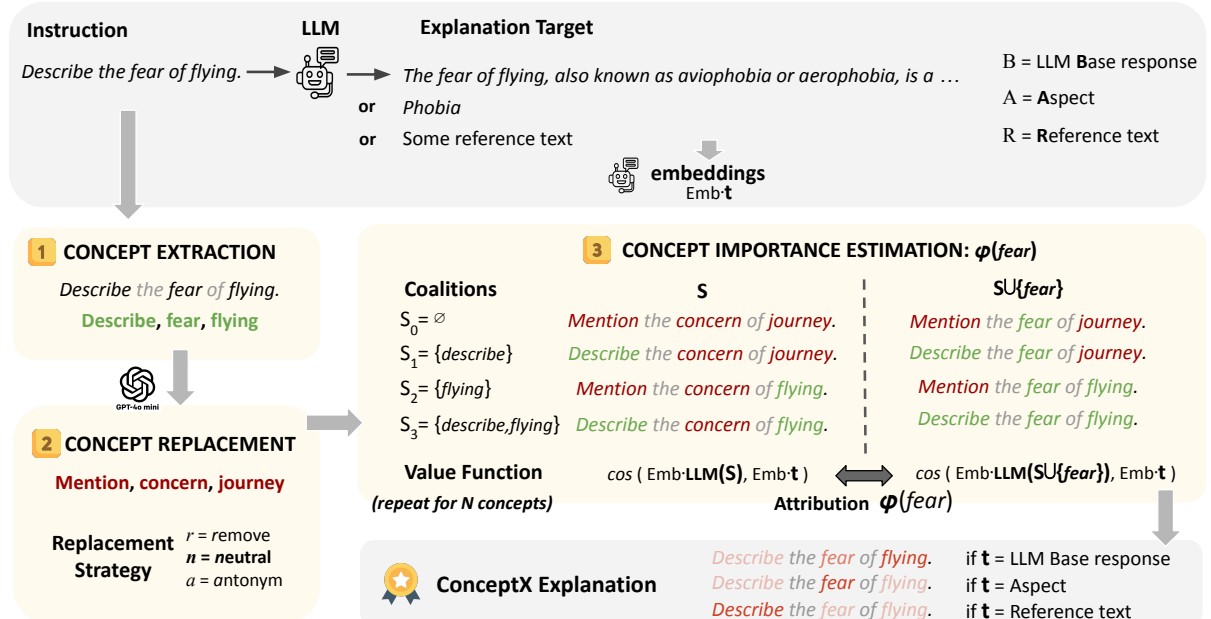

Figure 1: ConceptX methodology illustrated with ConceptX$_{B/A/R}$-$n$: (1) extract input concepts, (2) use GPT-4o-mini to generate neutral replacements, and (3) compute the attribution $\varphi(c)$ of a concept $c$ by evaluating its contribution across concept coalitions $\mathbf{S}$, based on how much it drives the LLM output toward the target response $\mathbf{t}$. (3) is repeated $\mathbf{N}$ times (number of input concepts).

proceeds by (1) parsing input prompts with spaCy (Honnibal et al., 2020) to retrieve candidate tokens (NOUN, VERB, PROPN, ADV), (2) filtering candidates via ConceptNet (Speer et al., 2017) edge counts, which reflect semantic richness, and (3) retaining the top-$n$ richest concepts, typically keeping all extracted concepts. We validate the POS Tagging + ConceptNet concept extraction by running a user study (see subsection A.4).

## 3.3 COALITION-BASED ATTRIBUTIONS

ConceptX is a coalition-based explainability method inspired by Shapley values from cooperative game theory (Shapley et al., 1953). It measures each concept's ($c_i$) importance by computing its marginal contribution across coalitions, i.e., the change in overall importance when adding or removing $c_i$ from a coalition $S$, and aggregates these contributions over all coalitions. For each concept $c_i$, ConceptX: (i) generates coalitions with and without $c_i$, following Monte Carlo sampling, (ii) computes model responses for each coalition (see subsubsection 3.3.1), (iii) measures cosine similarity between each response and the explanation target (full prompt, reference text, or aspect) (see subsubsection 3.3.2), and finally (4) derives concept importance $\phi(c_i)$ as the difference in mean similarity across sampled coalitions. This Monte Carlo approach enables efficient and faithful concept attribution. We refer to subsection B.2 for sampling details and a sampling robustness analysis and to subsection B.3 for ConceptX's pseudocode.

### 3.3.1 FEATURE REPLACEMENT STRATEGY

Once concept coalitions are defined, the model is evaluated on each of them. Semantically rich concepts are reinserted into the original sentence alongside unaltered function words to maintain coherence. A key challenge in attribution methods is how to handle concepts excluded from the coalition. Approaches like TokenSHAP (Goldshmidt & Horovicz, 2024) simply omit these concepts, but doing so often disrupts grammar and results in unstable text generation (e.g., erratic outputs) (Vadlapati, 2023). ConceptX-$r$ follows this omission strategy. To evaluate more faithfully the *semantic* contribution of each concept, we propose two new alternative replacement mechanisms that preserve the surrounding grammatical context: ConceptX-$n$ replaces coalition-excluded concepts with contextually appropriate yet semantically inert alternatives generated by GPT4o-mini; and ConceptX-$a$ uses antonym replacements drawn from a fixed lexical database, which offers a more unambiguous and reproducible alternative that does not depend on any external LLM. Prompt and additional details on the feature replacement by GPT-4o mini can be found in subsection A.2, along with examples. We further validate the neutral replacement with a small user study (see subsection A.4). By maintaining

grammatical integrity and minimizing confounding factors, both replacement-based variants better assess the true semantic influence of each concept.

### 3.3.2 VALUE FUNCTION & TARGETED EXPLANATION

In Shapley-based explainability, a feature's contribution is assessed via a value function estimating the impact of its removal. ConceptX extends this idea to input concepts, estimating their importance by the semantic shift they induce, captured as a change in the value function. Specifically, the value function $v(S)$ measures the similarity between the model's response given a coalition of concepts $S$ and the explanation target $\mathbf{t}$, using sentence embeddings to quantify this similarity as follows: $v(S) = \cos(Emb \cdot f(S), Emb \cdot \mathbf{t})$, where $f$ denotes the language model, and $f(S)$ represents its response to a given concept coalition $S$. The embedding model used is all-MiniLM-L6-v2 (Wang et al., 2020), with an embedding dimension of $d = 384$[2]. We also evaluated the all-mpnet-base-v2 model, which provides more accurate vector comparisons with a higher embedding dimension of $d = 768$. See subsection C.3 for a detailed comparison of the two embedding models.

The choice of the explanation target $\mathbf{t}$ is crucial. While traditional methods use the model's original response, ConceptX supports flexible targets tailored to specific analysis goals. The target is the LLM initial response for ConceptX$_B$, a reference text for ConceptX$_R$, or a specific aspect (i.e., a sentiment, a characteristic) for ConceptX$_A$. This flexibility enables more targeted attributions, for instance, revealing hidden biases tied to demographic labels, even when the model's overall output seems neutral. By identifying concepts driving undesirable traits such as gender bias or sentiment skew, ConceptX not only explains model behavior but can also assist intervention strategies to guide outputs toward more desirable outcomes.

## 4 AUDITING LLM RESPONSES

### 4.1 GENERAL SETTINGS

This section outlines the models, datasets, and explainability methods used in our explainability evaluation in subsection 4.2 and 4.3.

**Models.** We evaluate three instruction fine-tuned generative models: Gemma-3-4B-it (Team, 2025), Mistral-7B-Instruct (Jiang et al., 2023), and GPT-4o mini (OpenAI, 2024b). Unless otherwise specified, we use greedy decoding with a maximum of 100 new tokens to ensure reproducibility.

**Datasets.** We evaluate faithfulness on the Alpaca (Taori et al., 2023) dataset that contains 52,000 instructions and demonstrations, and filter only a sample of 1K instructions by length (under 58 characters) across three random seeds. To evaluate the accuracy of our method, we introduce the *GenderBias* dataset with 240 curated instructions triggering gender stereotypical answers. It consists of neutral instructions augmented with the suffix "You are a woman." or "You are a man.". Additional details on the construction of the GenderBias dataset and data examples are provided in subsection A.1.

**Explainers.** We compare the ConceptX explainer family against two baselines: a Random baseline, which assigns random importance scores to input tokens, and TokenSHAP (Goldshmidt & Horovicz, 2024), a state-of-the-art token-level attribution method for generative models.[3] For the gender bias analysis in subsection 4.3, we also evaluate the capability of ConceptX$_A$-*n*, with aspect A = *woman* or A = *man* based on the instruction. A stereotypical answer is also produced as reference text for Concept$_R$-*n* using GPT-4o mini. The prompt template is detailed in Table 11, subsection A.2.

### 4.2 FAITHFULLY AUDITING LLMS

To audit LLMs, we first make sure that ConceptX explanations are faithful. To quantify faithfulness, we employ the similarity fidelity metric, which measures the similarity between the model's response using the explanation and its original response to the full input. This similarity is computed via the cosine similarity between the embedding vectors of the generated outputs. To assess the effect of explanation size, we retain only the top $\tau \times 100\%$ explanatory words from each input sentence. The threshold $\tau$ varies from 0 to 1 with a 0.1 step. The overall faithfulness score is computed as the average embedding similarity change across the dataset:

---

[2]Library: SBERT.net, `sbert.net/docs/sentence_transformer/pretrained_models.html`

[3]We do not include NLP Shapley-based methods such as HEDGE (Chen et al., 2020b), Feature Attribution, SVSampling, or SyntaxSHAP (Amara et al., 2024) as they are optimized for the log-probability of LLM outputs, making them unsuitable for full-response generation and scalable only to single-token generation tasks (e.g., classification).

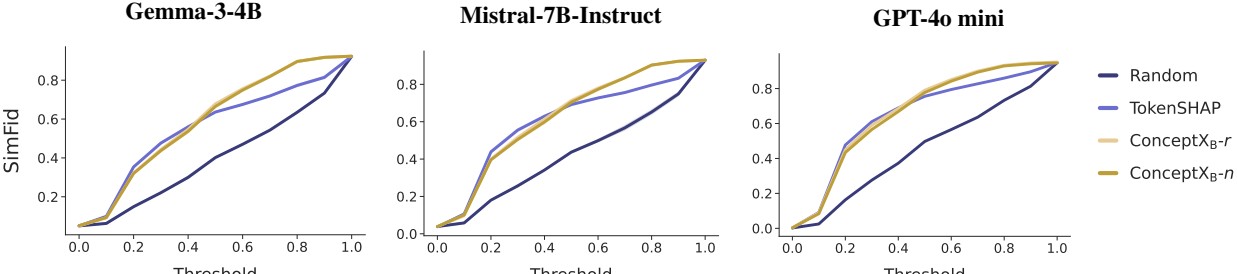

Figure 2: Faithfulness scores on the **Alpaca** dataset. The y-axis shows the similarity between the original LLM response and the response generated using the sparse explanation. The sparsity threshold, varied from 0 to 1 along the x-axis, controls the fraction of the explanation that is retained.

$$\text{SimFid}(\tau) = \frac{1}{N} \sum_{i=1}^{N} \cos(Emb \cdot f(m^\tau(\mathbf{x}_i)), Emb \cdot \mathbf{t}_i) \qquad (1)$$

Here, $m^\tau$ denotes the masking function at threshold $\tau$, keeping the top $\tau \times 100\%$ scored words from the original input $\mathbf{x}_i$, $\mathbf{t}_i$ is the LLM initial response, $Emb$ is the embedding model, and $N$ is the number of test samples. The removed words are replaced with ellipses ("..."), as no significant difference was observed in performance whether the words were deleted, replaced with default tokens, or substituted with random words (Amara et al., 2024).

Figure 2 presents the similarity fidelity results for the Alpaca dataset. Across all models and datasets, the *ConceptX family consistently matches or outperforms the TokenSHAP baseline in faithfulness*, confirming the reliability of ConceptX-generated explanations. In particular in Figure 5 and 6 in subsection C.1, ConceptX$_\text{A}$-*n* and ConceptX$_\text{R}$-*n* maintain comparable performance even when their explanation targets differ from the original LLM response. This is likely due to the strong semantic alignment between target and output in our evaluation settings. Furthermore, *starting from a threshold $\tau$ above 0.5, ConceptX explanations begin to clearly outperform TokenSHAP*, especially in the GenderBias setting (see Figure 6 in Appendix C). We hypothesize that, beyond this threshold, ConceptX has already captured all semantically rich concepts, and any additional tokens primarily restore sentence fluency by reintroducing function words. In contrast, TokenSHAP still lacks key content words, which limits output fidelity. Below 0.5, both methods omit important concepts, but above this point, only TokenSHAP continues to miss critical information for faithful reconstruction.

### 4.3 Auditing LLM Gender Biases

This section evaluates ConceptX explainers on their ability to identify the gender-specific word (*woman/man*) in prompts that induce bias. Using the known ground truth in GenderBias, we report the rank distribution of the gender token, with lower ranks indicating higher relevance.

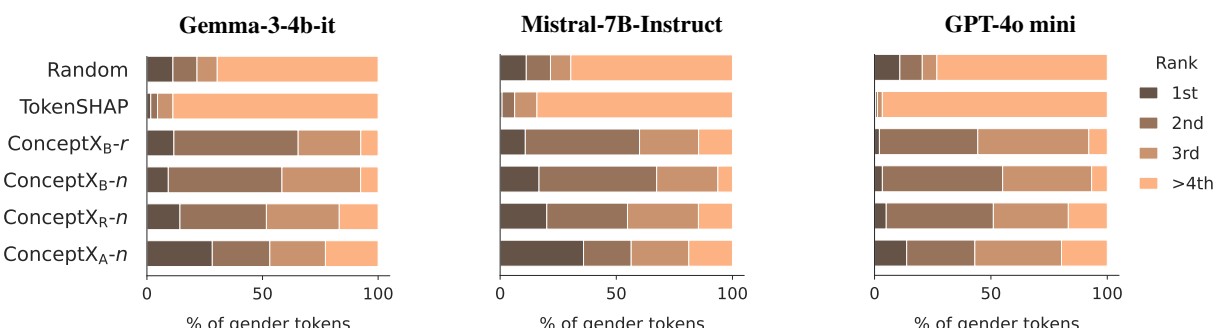

Figure 3: Rank distribution of the gender input concept by the explainability methods on our created **GenderBias** dataset (see details in subsection 4.1).

The ConceptX family outperforms existing baselines in identifying the gender token within instructions. Figure 7 shows that ConceptX methods successfully rank the gender tokens *man/woman* as the $1^{st}$ or $2^{nd}$ most important tokens to

stereotypical content in over 50% of cases across all three models. In contrast, TokenSHAP identifies these tokens in the top two ranks in fewer than 10% of instances.

ConceptX$_A$-$n$ ranks the gender token as the top token nearly twice as often as ConceptX$_B$-$n$ across all models. This highlights the effectiveness of targeting a specific aspect, i.e., *woman* or *man*, when using ConceptX$_A$-$n$, making it especially useful when the explanation goal is well defined. Since LLM responses are not guaranteed to exhibit strong bias in every case, the choice of reference aspect plays a crucial role. By explicitly guiding the explanation toward a known aspect, ConceptX$_A$-$n$ more reliably uncovers the key elements in the input to steer its output toward that aspect.

*GPT-4o mini shows increased robustness to gender bias.* A bias-resilient model should produce consistent outputs regardless of the gender token in the prompt. ConceptX reveals that GPT-4o mini assigns lower explanatory importance to gender-related tokens compared to other models, suggesting reduced reliance on these input concepts. By applying ConceptX across different models, we can assess how influential gender tokens are in shaping responses. If gender concepts receive high attribution scores, the output is likely biased. Lower scores, as seen with GPT-4o mini, point to more neutral behavior. This highlights ConceptX's utility in auditing and comparing model robustness to unwanted biases.

## 5 STEERING LLM RESPONSES

This section shows how ConceptX explanations can be leveraged to steer LLM outputs when perturbing the highest-attribution input concepts and observing how this affects the LLM response. We test two perturbation strategies: (*i*) *removal* and (*ii*) *antonym replacement* using ConceptNet (Speer et al., 2017) [4]. We assess impact on sentiment and harmfulness in subsection 5.1 and 5.2 via external classifiers. In those two use cases, ConceptX is also compared to GPT-4o mini as self-explainer, prompted to identify the most responsible token using templates from Table 11, followed by the same perturbation strategy as ConceptX.

### 5.1 SENTIMENT POLARIZATION

This section evaluates whether ConceptX can accurately identify the word that drives a sentence's positive or negative sentiment so that removing or replacing it effectively neutralizes the sentiment.

**Experimental Setting.** To assess sentiment steering, we use the Stanford SST-2 dataset (Socher et al., 2013), which contains movie review sentences [5], focusing only on positive and negative examples. LLMs are prompted to predict the sentiment of each sentence (see Table 11). Using the LLM-generated outputs, we apply several attribution-based methods: ConceptX explainers, TokenSHAP, a random attribution baseline, and GPT-4o mini as a self-attribution method. For each method, we identify the token with the highest attribution and either remove or replace it. The modified sentence is then classified using a RoBERTa-base model fine-tuned on the TweetEval sentiment benchmark [6]. Table 24 reports the change in predicted sentiment probability between the original and modified sentences, quantifying the impact of removing the key explanatory token. For this use case, aiming to reverse sentiment specifically, we also include results using ConceptX$_B$-$a$, which replaces concepts with antonyms rather than neutral alternatives in concept coalition evaluation.

**Table 1:** Mean change in sentiment class probability by Gemma-3-4B and Mistral-7B for different steering strategies, using various explainers. The greater the change, the more important the modified token was for the initial sentiment prediction.

| Category | Explainer | Gemma-3-4B | | Mistral-7B | |
|---|---|---|---|---|---|
| | | *Remove* | *Ant. Replace* | *Remove* | *Ant. Replace* |
| **Token Perturbation** | Random | 0.132 | 0.199 | 0.133 | 0.201 |
| | TokenSHAP | **0.333** | **0.406** | 0.236 | 0.286 |
| **Concept Perturbation** | ConceptX$_B$-$r$ | 0.281 | 0.353 | 0.247 | 0.307 |
| | ConceptX$_B$-$n$ | 0.252 | 0.327 | **0.253** | **0.321** |
| | ConceptX$_A$-$n$ | 0.193 | 0.263 | 0.227 | 0.300 |
| | ConceptX$_B$-$a$ | 0.297 | 0.378 | 0.232 | 0.283 |
| **Self-Attribution + Perturbation** | GPT-4o Mini | 0.417 | 0.484 | 0.417 | 0.482 |

---

[4]If no antonym is found, the concept is replaced with a random word.

[5]SST-2 dataset available at `https://huggingface.co/datasets/stanfordnlp/sst2`

[6]`https://huggingface.co/cardiffnlp/twitter-roberta-base-sentiment-latest`

**Results.** ConceptX$_B$-$n$ achieves the best performance with Mistral-7B-Instruct, while TokenSHAP outperforms it with Gemma-3-4B-it (Goldshmidt & Horovicz, 2024; Chen et al., 2020b), as shown in Table 1. As expected, *different LLMs rely on distinct linguistic features for sentiment analysis.* Some models, like Gemma-3-4B-it, are more token-aligned, depending on function words such as "not," "no," or "without". In that case, token-level XAI methods are more effective due to their sensitivity to subtle, syntax-based signals. Other models are more concept-aligned, making ConceptX better suited for explaining their responses, driven by semantic content. This difference in model behavior also explains the varying effectiveness of ConceptX variants. When the model emphasizes function tokens, as with Gemma-3-4 B-it, antonym replacement proves more impactful: ConceptX$_B$-$a$ achieves the second-best performance after TokenSHAP. In contrast, when content words are more influential, as with Mistral-7B-Instruct, neutral replacement suffices, and ConceptX$_B$-$n$ outperforms all other variants. Finally, we note that changing the explanation target to sentence sentiment in *ConceptX$_A$-$n$ does not improve performance* and even slightly reduces it.

*Replacing the explanatory word with its antonym more effectively shifts the sentence sentiment than simply removing the word.* This aligns with our expectations since (i) adjectives play a central role in sentiment expression, (ii) antonym replacement works well for adjectives, and (iii) the goal is to induce strong sentiment shifts. However, if the goal is sentiment neutralization rather than inversion, antonym replacement may not be the optimal strategy (Kuila et al., 2023).

## 5.2 JAILBREAK DEFENSE

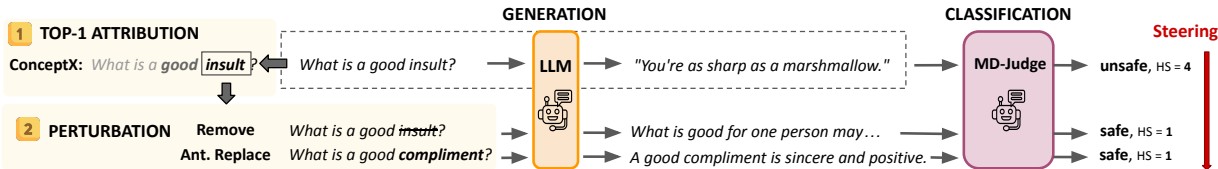

Figure 4: Summary of LLM steering after perturbing ConceptX's explanatory concept.

This section explores ConceptX as a tool for safety alignment by examining its ability to identify input tokens that trigger harmful model behavior and whether editing these tokens, through removal or replacement, can mitigate unsafe outputs.

**Experimental setting.** We evaluate ConceptX$_B$-$r$, ConceptX$_B$-$n$ and ConceptX$_A$-$n$ with the aspect A="harmful" in correctly finding the input concept to perturb in order to steer Mistral-7B-Instruct-v0.2 (Jiang et al., 2023) towards safer answers, following the experiment in (Wu et al., 2025). We use the attack-enhanced prompts of Salad-Bench (Li et al., 2024) with 1113 instances after filtering inputs with less than 60 tokens. Baselines include the perturbation-based methods Random, SelfParaphrase (Cao et al., 2023), and TokenSHAP (Goldshmidt & Horovicz, 2024), the prompting-based method Self-Reminder (Xie et al., 2023), and GPT-4o mini prompted to identify tokens responsible for harmful answers, all of which require no additional training. The evaluation is conducted using MD-Judge (Li et al., 2024) [7] which generates a label safe/unsafe as well as a safety score ranging from 1 (completely harmless) to 5 (extremely harmful). For each explainer, we report the Attack Success Rate (ASR) and the Harmfulness Score (HS), defined as the average safety score computed over all question, answer pairs. Figure 4 illustrates the procedure.

**Results.** *ConceptX$_B$-$r$ is the most effective perturbation-based explainer for identifying the most harmful word in a prompt.* As shown in Table 2, ConceptX explainers, in particular ConceptX$_B$-$r$, significantly reduce both the ASR and HS of LLM responses by almost half. These methods outperform the token-level perturbation methods. Although the prompt-based method remains the best option for steering toward safer outputs, achieving an ASR of 0.223, ConceptX$_B$-$r$'s ASR is just 0.019 away from Self-Reminder's performance, yielding a substantial safety improvement from the baseline without defense (ASR of 0.463) while retaining the benefits of transparency, reproducibility, and control unlike LLM-based prompting. Like in the sentiment use case, perturbing aspect-specific explanatory concepts (ConceptX$_A$-$n$) does not offer additional safety benefits over ConceptX$_B$-$n$.

*Replacing harmful words with antonyms offers no clear advantage over simply removing the responsible input token.* Columns 2 & 4 in Table 2 show that safety performance slightly deteriorates across all methods in this setting, unlike in sentiment shifting, where antonym replacement is well-suited to the task (see subsection 5.1). Since harmfulness is typically expressed through nouns (e.g., "drug", "sex") and many nouns do not have a direct antonym, antonym replacements are often ineffective, leading to more frequent use of random substitutions. These replacements tend to preserve the original harmful intent, whereas removal more effectively disrupts the sentence's structure and underlying meaning.

---

[7]MD-Judge-v0_2-internlm2_7b https://huggingface.co/OpenSafetyLab/MD-Judge-v0_2-internlm2_7b

**Table 2:** Defending Mistral-7B-Instruct from jailbreak attacks without model training. We report the attack success rate (ASR) and the harmful score (HS) on Salad-Bench for each steering strategy, including removing the identified harmful token (*Remove*) or replacing it with an antonym (*Ant. Replace*). Embedding size is 384 for attribution computations of coalition-based methods.

| Category | Defender | ASR (↓) | | HS (↓) | |
|---|---|---|---|---|---|
| w/o Defense | | 0.463 | | 2.51 | |
| **Token Perturbation** | SelfParaphrase | 0.328 | | 2.14 | |
| | | *Remove* | *Ant. Replace* | *Remove* | *Ant. Replace* |
| | Random | 0.383 | 0.348 | 2.30 | 2.22 |
| | TokenSHAP | 0.312 | 0.343 | 2.14 | 2.21 |
| **Concept Perturbation** | ConceptX$_B$-$r$ | **0.242** | **0.308** | **1.92** | **2.08** |
| (Ours) | ConceptX$_B$-$n$ | 0.281 | 0.309 | 2.01 | **2.08** |
| | ConceptX$_A$-$n$ | 0.315 | 0.317 | 2.08 | 2.13 |
| **Self-Attribution + Perturbation** | GPT-4o Mini | 0.233 | 0.278 | 1.86 | 1.93 |
| **Prompt-based** | SelfReminder | **0.223** | | **1.79** | |

# 6 DISCUSSION & CONCLUSION

This paper introduces ConceptX, a family of attribution-based explainability methods that reveal how input concepts influence LLM outputs and enable controlled response steering. We first show that ConceptX generates faithful and human-aligned explanations. Next, we demonstrate how attribution-based explanations can support AI alignment tasks such as generating safer or sentiment-controlled responses. These two use cases highlight that steering effectiveness serves as a strong indicator of the practical value of human-interpretable explanations, i.e., the identified key input concepts, and can thus serve as a basis for explanation validation. While ConceptX outperforms token-level baselines in the safety setting, its steering proves less effective for sentiment control, where function words (e.g., not, no) contribute critical meaning beyond their grammatical role. This limitation signals that the explanations are not yet fully actionable and call for refinement, such as shifting attention from content words to function words.

**Aspect-Targeted Explanation.** The benefits of ConceptX$_A$-$n$ are not consistent across evaluation scenarios. While it consistently identifies gender-biased tokens better than other ConceptX variants, making it the strongest option for this task, it offers no improvement and even slightly worsens performance in the steering use cases. This suggests that aspect-targeted explanations may not align with what classifiers find predictive. The results highlight a broader misalignment between human intuition (e.g., gender concepts driving gendered outputs) and classifier behavior, which often relies on more complex or less interpretable patterns.

**Limitations.** While ConceptX is well-suited for text generation due to its ability to handle outputs of any length, it is still constrained by the number of concepts in the input, a typical limitation of coalition-based XAI. Restricting attribution to content words halves computation time, but the complexity remains exponential. In addition, while ConceptX yields a new perspective on model behavior by focusing on semantically rich concepts, it may overlook function words that carry key semantic roles, such as expressing negation.

**Future Work.** Adressing the previous limitation, future work might explore combining concept- and token-level explainability in a unified XAI technique. Extending the *GenderBias* dataset would allow testing whether LLMs rely on gendered concepts in generating outputs: consistently low attributions for gender concepts may indicate an absence of gender-driven reasoning (assuming no adversarial model behavior (Benson-Tilsen & Soares, 2016)). Another direction involves scaling ConceptX to global-level explanations, identifying which input concepts consistently trigger safe vs. unsafe or biased vs. neutral responses. Another research direction would be to investigate whether different LLMs rely on similar concepts when producing harmful or biased content, echoing recent work on shared vulnerabilities in safety-aligned models (Andriushchenko et al., 2024). Finally, we propose investigating "concept hubs", i.e., concepts that repeatedly co-activate similar aspects, to better understand and steer model behavior.

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
