# OpenReview forum: "Concept-Level Explainability for Auditing & Steering LLM Responses"
_ICLR.cc/2026/Conference — ICLR 2026 Conference Withdrawn Submission_

### Official Review · Reviewer_Hwbo · 2025-10-27

**Soundness:** 2
**Presentation:** 2
**Contribution:** 2
**Rating:** 2
**Confidence:** 5

**Summary:**

ConceptX is a model-agnostic, concept-level attribution method: it extracts semantically rich input concepts (via ConceptNet + POS), estimates their influence with Shapley-style coalitions, and preserves fluency via neutral/antonym replacements. It uses embedding similarity to a target (original output, a reference, or an aspect) to score concepts, and shows wins over TokenSHAP in faithfulness, bias auditing, and some steering tasks (sentiment, jailbreak defense).

**Strengths:**

* Semantics-first explanations: focuses on content concepts (ConceptNet + POS) and targets meaning, not token overlap.

* Fluent interventions & flexible targets: in-place neutral/antonym replacements and B/R/A targets (output, reference, aspect) make explanations actionable.

* Useful for auditing/steering: improves faithfulness on Alpaca; identifies gender tokens better than TokenSHAP; reduces jailbreak ASR and harmfulness.

**Weaknesses:**

(1) Because GPT-4o-mini both writes the “neutral” edits and similar LLMs judge them, the test mostly checks how consistent these models are with themselves—not which words truly matter. And when you aim at an aspect like “harmful,” you’re really using GPT-4o-mini’s own built-in idea of what “harmful” means.


(2)  The method leans on high-dimensional cosine where distinctions are unstable. But does cosine-similarity of sentence embedding

 (3) Performance is mixed across models (e.g., worse on Gemma-3-4B, better on Mistral-7B) although explained but this makes the method less reliable across different llms. The GenderBias set (n=240) is too small for significance and post-hoc claims (token- vs concept-aligned) are unvalidated, suggesting limited robustness. Please perform ablation studies to make the method more reliable.

(4) The Monte Carlo sampling for Shapley values requires exponentially many samples (2^n coalitions), but the paper doesn't specify sampling rates or provide convergence guarantees

**Questions:**

Same as weakness

---

### Official Review · Reviewer_yHC2 · 2025-10-29

**Soundness:** 2
**Presentation:** 2
**Contribution:** 2
**Rating:** 2
**Confidence:** 4

**Summary:**

ConceptX is a model-agnostic, concept-level attribution method for text generation. It selects semantically rich input tokens (via POS + ConceptNet), estimates their importance with coalition/Shapley-style sampling, and uses embedding-based cosine similarity to target explanations (to the base output, a reference text, or a semantic “aspect” like gender bias or harmfulness). It also proposes in-place neutral/antonym replacements to preserve fluency when probing concepts, and shows gains in faithfulness, bias auditing, and prompt-level steering vs. TokenSHAP across several LLMs.

**Strengths:**

Targets meaning via sentence-embedding similarity, enabling aspect-focused explanations (bias/sentiment/harm).

Fluent probing: Neutral/antonym replacements reduce grammar breakage vs. token deletion, improving stability.

Practical wins: Better gender-token ranking and strong jailbreak-defense steering without retraining.

**Weaknesses:**

* The value function relies on cosine similarity in a sentence-embedding space; the paper provides no theoretical justification that this similarity measures concept contribution, and its faithfulness metric reuses the same embedding-cosine, making validation partly circular.

* ConceptX averages marginal effects over random coalitions (“Shapley-inspired”) but doesn’t check whether the Shapley axioms are satisfied. In practice, that means the individual concept scores may not add up to the overall effect, making the attribution internally inconsistent.

* The authors have said this method as model-agonistic but in the experiement section the authors have told: "As expected, different LLMs
rely on distinct linguistic features for sentiment analysis. Some models, like Gemma-3-4B-it, are more token-aligned,
depending on function words such as "not," "no," or "without". In that case, token-level XAI methods are more effective
due to their sensitivity to subtle, syntax-based signals. Other models are more concept-aligned, making ConceptX better
suited for explaining their responses, driven by semantic content. This difference in model behavior also explains the
varying effectiveness of ConceptX variants. "
This tells the method is not not model-agonistic but depends on how the model works.

**Questions:**

* I request the authors to compare the method with the recent attribution methods as the paper lacks comparisons. Also ablation studies on the method will increase the faithfulness of the method.

* Also refer the weakness as questions please.

---

### Official Review · Reviewer_GXjJ · 2025-10-31

**Soundness:** 3
**Presentation:** 3
**Contribution:** 2
**Rating:** 4
**Confidence:** 4

**Summary:**

This research tries to study the input attribution problem for LLMs. The authors propose a pipeline to first extract meaningful words from the input texts, then replace them with other words by prompting an LLM. The modified inputs will be used to collect new output responses, and the importance of each replaced input word is measured by the cosine similarity between the embeddings of the new output and the original output.

**Strengths:**

1. The pipeline of manually replacing the input words and monitoring the changes in the semantic similarity between perturbed and original output responses faithfully aligns with the idea of input attribution.

2. The writing of this manuscript is clear.

3. A large number of experiments on using the explanations for downstream tasks are good.

**Weaknesses:**

1. The paper is missing an important reference [1], which has extended the input attribution methods on generative models. The authors should compare the proposed method with this baseline.

2. The authors claim that they are providing "concept-level" explanations, while they are just extracting single meaningful words, such as nouns, verbs, and so on. This is an overclaim statement.

3. The experiment part should also report the time that different methods took.


[1] Wu, Xuansheng, et al. "From Language Modeling to Instruction Following: Understanding the Behavior Shift in LLMs after Instruction Tuning." Proceedings of the 2024 Conference of the North American Chapter of the Association for Computational Linguistics: Human Language Technologies (Volume 1: Long Papers). 2024.

**Questions:**

Please see the Weaknesses.

---

### Official Review · Reviewer_uXqT · 2025-11-03

**Soundness:** 2
**Presentation:** 3
**Contribution:** 2
**Rating:** 4
**Confidence:** 3

**Summary:**

The paper introduces ConceptX, a model-agnostic, concept-level attribution method for generative LLMs. Instead of token-level importance, ConceptX extracts semantically rich content words using ConceptNet and POS tags, then estimates each concept’s contribution via a Shapley-style Monte Carlo coalition scheme. Crucially, it preserves grammaticality by replacing non-selected concepts (remove / neutral substitute/antonym) and scores coalitions with semantic similarity to a target: the base output, a reference text, or an aspect (e.g., “harmful”, “gendered”). The method is used for auditing (faithfulness; gender-bias identification) and steering (sentiment shift; jailbreak defense).

**Strengths:**

- The problem of explainability in terms of semantically meaningful words is significant for LLM research communities.
- Experimental gains: On Alpaca, ConceptX equals or beats TokenSHAP across models over a range of sparsity thresholds.

- No training is required, e.g., ConceptX notably reduces ASR from 0.463 to 0.242 and lowers harmfulness vs token-level and paraphrasing baselines (close to Self-Reminder).

**Weaknesses:**

- The definition of the concept is vague? Is the word with a high degree of richness in the ConceptNet a concept?
- Evaluation coupling to embedding choice: Value function and metrics hinge on sentence-embedding similarity; sensitivity to embedding model, domain, and target choice (B/R/A) could bias results. The appendix mentions comparisons, but stronger analysis (calibration/ablation) would help.
- Despite Monte Carlo sampling and concept filtering, coalition methods still carry exponential characteristics; runtime and budget implications for long prompts or many concepts are only partially addressed.
- Additional recent concept-level or SAE/activation steering baselines (or stronger prompting baselines tuned per task) could sharpen claims

**Questions:**

See cons.

---

### Note · Authors · 2025-11-18

I have read and agree with the venue's withdrawal policy on behalf of myself and my co-authors.